# Inverting cognitive models with machine learning to infer preferences from fixations

**Anonomyous Author(s)**

## Abstract

Inferring an individual's preferences from their observable behavior is a key step in the development of assistive decision-making technology. Although machine learning models such as neural networks could in principle be deployed toward this inference, a large amount of data is required to train such models. Here, we present an approach in which a cognitive model generates simulated data to augment limited human data. Using these data, we train a neural network to invert the model, making it possible to infer preferences from behavior. We show how this approach can be used to infer the value that people assign to food items from their eye movements when choosing between those items. We demonstrate first that neural networks can infer the latent preferences used by the model to generate simulated fixations, and second that simulated data can be beneficial in pretraining a network for predicting human-reported preferences from real fixations. Compared to inferring preferences from choice alone, this approach confers a slight improvement in predicting preferences and also allows prediction to take place prior to the choice being made. Overall, our results suggest that using a combination of neural networks and model-simulated training data is a promising approach for developing technology that infers human preferences.

**Keywords:** Fixation, Cognitive Models, Neural Networks, Inverse Reinforcement Learning

## 1. Introduction

Key to building systems that help people make better choices is inferring what people want from their behavior (Hadfield-Menell et al., 2016). How can this inference take place? Cognitive models, which specify how latent preferences generate behavior, could in principle be applied to this problem. By using Bayesian inference to invert such a model, we can infer preferences from behavior. However, cognitive models often fail to capture idiosyncratic relationships between preferences and behavior, and inverting such models is computationally burdensome. In contrast, machine learning models such as neural networks offer a way to make inference computationally feasible and have greater flexibility to capture arbitrary relationships. However, training such models requires vast amounts of behavioral data.

In this work, we propose and test a new solution to the problem of inferring preferences from behavior, combining the strengths of cognitive models and neural networks. Our approach is to satisfy the need for massive data to train neural networks by augmenting limited available real human data with simulated data from a cognitive model. We apply this approach to the problem of inferring human preferences over food items from visual fixations between those items made during the decision-making process. Our results demonstrate that neural networks are able to learn, from simulated data, to invert a computationally intensive cognitive model for how individuals decide where to fixate while making a decision given their preferences over items. Additionally, pretraining a network with simulated data and fine-tuning with limited human data allows prediction of people's self-reported preferences

from their fixations. This demonstrates a new approach for how cognitive models can be used to address key limitations of deploying neural networks in human-interaction systems.

In machine learning, the problem of inferring another agent's preferences has been cast as inverse reinforcement learning (IRL; Ng and Russell, 2000). IRL specifies a generative model whereby agents have latent preferences (formalized as a utility function over task states and/or actions) and make decisions that maximize those preferences. This generative model, relating preferences to behavior, is inverted to predict the maximum a posteriori (MAP) preferences that generated the observed behavior. This general framework of inferring preferences by inverting a decision model has also formed the basis of cognitive models for how individuals make inferences about others preferences based on their behavior (Lucas et al., 2014; Jern et al., 2017; Baker et al., 2017; Jara-Ettinger, 2019). Cognitive science has also recently provided more sophisticated models of how humans make decisions, which can provide more accurate models relating preferences to actions to guide inference (Ho and Griffiths, 2022), and can expand the observables over which inference can occur to data beyond choices (e.g. response times; Gates et al., 2021).

Although IRL defines how preference inference can occur in principle, its practical use has been limited by the computational challenge of inverting decision models. Finding the MAP preferences typically involves searching over, and computing the likelihood of, candidate utility functions. For many cognitive process models, computing this likelihood for a single utility function can be quite computationally intensive. This makes a full search process too computationally expensive to be deployed in real-time inference settings. As a step toward making inference faster, recent work has shown that it is possible to implement IRL in neural networks, for which inference is fast (Rabinowitz et al., 2018). However, this approach requires large amounts of labeled training data, which is often unavailable for real-life applications. Here, we test whether use of simulated data can alleviate this need for real human data.

Specifically, we consider the problem of predicting preferences from visual fixations, which is particularly valuable for virtual and augmented reality systems. When individuals make a choice between items to acquire, they tend to move their gaze between potential items in a stereotypic manner. This process has been studied experimentally in tasks where a participant is presented with a screen displaying snack items, and is required to select which of them they would prefer to eat at the end of the experiment (Krajbich et al., 2010; Krajbich and Rangel, 2011). Recent work suggests that when making such decisions, people fixate on the different options in a way that depends on independently provided ratings of how much they like those items (Anonymous; Gluth et al., 2018; Jang et al., 2021). These relationships in principle make it possible to predict individuals' utility over items from their fixations. Prior studies have found that, indeed, it is possible to use the total as well as proportion of time individuals spend fixating on different items to predict, to some extent, individuals' preferences for those items (Goyal et al., 2015; Glaholt et al., 2009).

Here, we examine how use of a cognitive model of how individuals select fixations can be applied to improve this inference. Anonymous presented a resource-rational model for how individuals select both where to fixate at any point in time, and when to stop fixating and make a choice in such tasks. According to the model, eye movements reflect optimally selected information-gathering computations that improve the participant's beliefs about the utilities of different snack items. These computations can lead to a better ultimate

decision, however they also incur a cognitive cost. By formalizing this process as a sequential decision problem (specifically, a meta-level Markov decision process), the optimal fixation policy can be identified. It was found that the sequences of fixations made by the optimal policy closely corresponded to participant's observed fixation behavior.

Our objective is to use simulated data from this model to train neural networks to infer an individual's preferences given their fixations. This work builds on work in cognitive science and machine learning that has combined neural networks with simulated data to either invert complex generative models or to predict human choices. For fitting cognitive models to behavior, recent work has used neural networks to approximate likelihood functions that might otherwise be intractable (Fengler et al., 2021). Closer to our application here is work that has trained neural networks to directly estimate mean parameters or sample from posterior distributions of complex models, by training networks with simulated data labeled with corresponding parameters (Radev et al., 2022; Papamakarios and Murray, 2016; Gonçalves et al., 2020; Ger et al., 2023; Yildirim et al., 2020). Neural networks used to predict human decisions have been pretrained with simulated data from cognitive models to make up for limited real human data (Bourgin et al., 2019). Finally, neural networks trained to predict human choices have in turn have been used to improve cognitive models through a process referred to as scientific regret minimization (Agrawal et al., 2020; Peterson et al., 2021; Kuperwajs et al., 2023).

We turn this approach toward the problem of estimating human preferences from eye fixations. Our specific approach is to train neural networks on simulated fixation and choice data from the model presented in Anonymous. We first test whether we can simply invert the model; that is, we provide neural networks with a sequence of simulated fixations followed by a choice and test whether they output correct utilities over the three items. Following this, we validate the approach using real human data on a trinary choice task, reported in Krajbich and Rangel (2011). We determine whether neural networks can predict people's reported utilities given their fixations and choices, how this compares to prediction using choice alone, and then also whether simulated data complements using human data alone in training models on this task.

## 2. Methods

### 2.1. Human Data

Human data consisted of 2965 trials reported in Krajbich and Rangel (2011) in which participants made choices over three food items after having the opportunity to engage in a sequence of fixations between them. Fixations, $f_{jt_i}$, reflect the item most fixated on in a .1 second bin. Prior to all choices, participants provided liking ratings (utilities) over the full set of items.

### 2.2. Simulated Data

Simulated data was generated using the model described in Anonymous. To simulate a single trial, $j$, a utility, $u_{js}$, was drawn for each snack item, $s$, from $P(u)$, which was defined by fitting a Gaussian distribution to the full set of item ratings from Krajbich and Rangel (2011). Given such "true" utilities over items, the model generates a sequence of fixations,

$f_{jt_i<T}$, over by items, followed by a choice, $c_{jt_i}$, $x_j = (f_{jt_1}, f_{jt_2}, ..., c_{jt_T})$. At a high level, each simulated fixation on item $s$ collects a sample from a distribution of item utilities centered on $u_{js}$, with Gaussian noise. This sample is used to increase the accuracy of an estimate of that item's utility. Optimal fixations reflect the information gathering actions that balance the benefit of making a choice with a more accurate utility estimates with the cost of spending additional time. Using this model, we simulated 1.8 million trials.

### 2.3. Input and Target Data Representation

For each trial, $j$, consisting of $T$ time-points, we represented that trial's sequence of fixations followed by a choice as a length-$T$ sequence of 6-length vectors, $x_{ji}$, for each time-point, $i = 1 : T$. For each time-point, $i < T$, the first 3 elements of $x_{ji}$ designated which of the 3 food items was fixated on at that time-point. The last 3 elements, which were active only for the final time-point, $T$, designated which of the three items was chosen on that time-point. Sequence-based models were trained to make a prediction of each of the three item's utilities at each time-point, $i$, in the sequence, using all input data up to time-point $i$. The target sequence thus consisted of a length-3 vector where each element, $j = 1 : 3$ contained the true utility of item $j$, $u_j$, repeated for each time-point in the sequence.

We compared models trained on both fixations and choice to a model trained on choice alone. For the model trained on choice alone, we trained a model that simply estimated two parameters reflecting the respective utilities of the chosen item and non-chosen items. We also defined a set of control models based on features that previous work has identified as predictive of preferences: the cumulative total and proportion fixation time on each item (Goyal et al., 2015; Glaholt et al., 2009). For each time point, we defined a length-9 vector, with three values indicating the current fixated item, three indicating the total fixation time on each item, and three indicating the proportion fixation time on each item. We then trained multi-layer perceptrons to map these features at each time-point to utility estimates for each item. Models including all of these features predicted utilities better than models using only a subset of them (Appendix Fig. 4). Thus, we used this as a control model for comparison to our approach.

### 2.4. Training Procedure and Hyperparameter Selection

Both simulated and human data were split into training (60%), validation (20%) and testing (20%) sets, which were used to respectively train the model, select hyperparameters, and test final accuracy. We trained LSTMs (Hochreiter and Schmidhuber, 1997), GRUs (Cho et al., 2014), and Transformers (Vaswani et al., 2017). Because qualitative results were the same across architectures, we show only LSTM results in the main text and present results for all networks in the appendix (Appendix Figs. 5-7). Control models used multi-layer perceptrons (MLPs). All networks were implemented in the Python package, PyTorch (Paszke et al., 2017). We used the Adam optimizer to identify network parameters that minimized the mean square error in predicting the set of training sequences. All training used a batch size of 32. For each task, for all networks, we used a grid search to identify the number of hidden units (and embedding dimensionality for transformers; out of $[8, 16, 32, 62, 128, 256, 512]$) and learning rate (out of $[.00001, .0001, .001]$). For each combination of these hyperparameters, we trained 5 models, each with different starting weights,

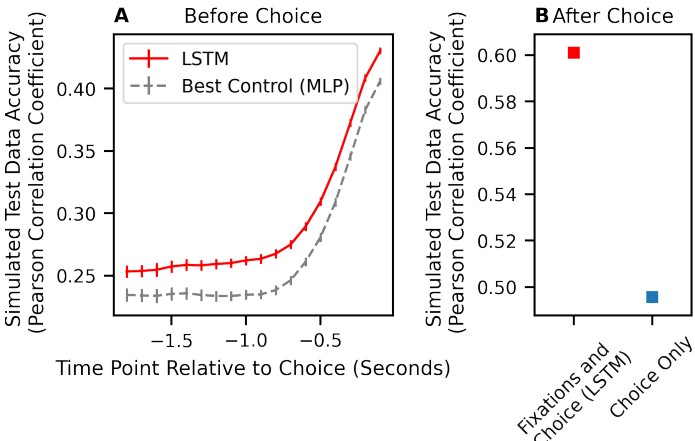

Figure 1: Results of training model on simulated data and testing on held-out simulated data. A. Predictive accuracy of neural networks at predicting simulated data utilities, at each time-point prior to a choice being made. The best control model used an MLP to map current item identity and both sum and proportion of fixation items up to a time-point to predict utility of all three items. B. Predictive accuracy after the choice is made. LSTMs trained on simulated fixation and choice data outperform a model which only uses the choice that was made.

for 2 million training sequences. We then averaged these 5 error-vs-training-number curves, smoothed them with a Gaussian kernel ($\sigma = 200$ batches) and selected the hyperparameters and number of training sequences that achieved minimum mean squared error. Note that when pretraining with simulated data, we use 1 million training examples of simulated data and then allow the number of human fine-tuning examples to vary (up to 2 million). For the transformer networks, we set the number of attention heads to 4 and the number of layers to 2. All other parameters were set to pytorch default values. All final results reflect using these hyperparameters and number of training sequences, averaged over 180 runs, each with randomized training data ordering and initial weights.

## 3. Results

### 3.1. LSTMs trained on simulated data can predict latent utilities

We first examined the ability of LSTMs trained on simulated fixation and choice data to predict corresponding latent utilities used to generate that data. An advantage for predicting utilities from fixations in addition to choices, as opposed to predicting from choices alone, is that prediction from fixations can be made prior to the choice occurring. Indeed, LSTMs were able to predict latent utilities at time-points prior to choice occurrence, from fixations alone, with prediction accuracy increasing up until the time of choice occurance (Fig. 1A). This prediction accuracy prior to choice outperformed a variety of control models, which used multi-layer perceptrons to map hand-designed features at a single-time point to predic-

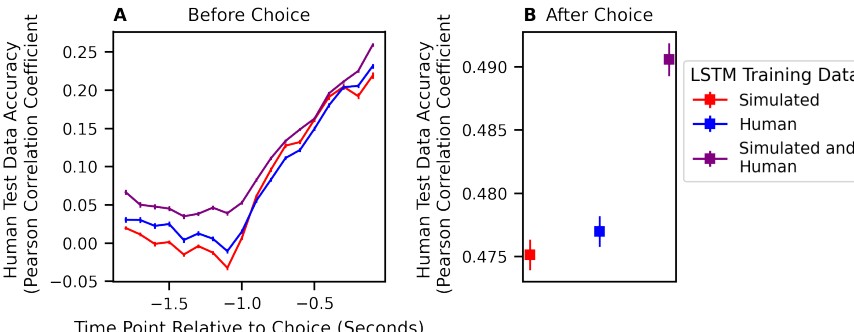

Figure 2: Testing on human data under different training regimes. LSTMs were trained using either simulated data alone, human data alone, or were pretrained with simulated data and fintuned with human data. Networks trained with both simulated and human data outperformed networks trained with either alone. A. Predictive accuracy of neural networks at predicting self-reported human item utilities, at each time-point prior to a choice being made. B. Predictive accuracy of neural networks at predicting utilities of human data after choice is made, using both fixation and choice information.

tion of utilities (see Methods). The best performing control model was provided the current fixation identity, the sum of fixations on each item up to that time-point, and the proportion of fixations up to that time-point (Appendix Fig. 4A). This control model achieved worse prediction accuracy than the LTSM model (independent sample t-test comparing accuracy correlations aggregated across time-points, $t(358) = 22.1, p < .001$) demonstrating that the LSTM can learn non-trivial sequential aspects of the relationship between fixations and preferences in the simulated data. LSTMs trained on fixations in addition to choice also conferred an advantage in predicting preferences after a choice was made compared to predictions made using choice alone (Fig. 1B; $t(358) = 92.6, p < .001$). This demonstrates an ability to learn about relationships between fixations and preferences in simulated data beyond just predicting which item will be chosen.

### 3.2. Simulated data complements human data in predicting human utilities

We next sought to examine the ability of LSTMs trained on fixation and choice data to predict human self-reported utilities of items from fixations and choices over those items. Additionally, we sought to determine whether training networks with simulated data provided a benefit over training with human data alone. We thus compared the ability of different LSTMs to predict real human self-reported preferences, varying whether the LSTMs were trained using simulated data only, human data only, or pretrained on simulated data and fine-tuned using human data. Networks pretrained on simulated data and finetuned with human data outperformed networks trained using either simulated data or human data alone, both when predicting preferences prior to a choice being made (Fig. 2A; Simulated and Human vs Simulated Only: $t(358) = 18.1, p < .001$; Simulated and Human vs Human

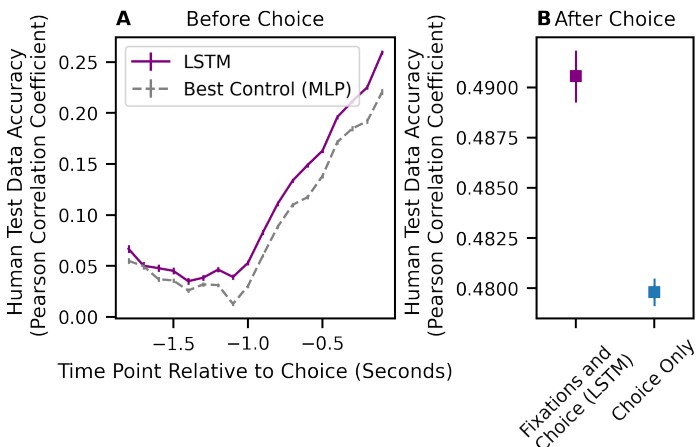

Figure 3: Comparison of LSTMs trained on simulated and human data to control model utilizing hand-crafted features and also model which uses choice alone. A. Predictive accuracy of neural networks at predicting self-reported human item utilities, at each time-point prior to a choice being made. The best control model used an MLP to map current item identity and both sum and proportion of fixation items up to a time-point to predict utility of all three items. B. Predictive accuracy on human data after the choice is made. LSTMs trained on simulated and human fixation and choice data outperform a model which only uses the choice that was made.

Only: $t(358) = 15.9, p < .001$) and also when predicting with knowledge of the choice (Fig. 2B; Simulated and Human vs Simulated Only: $t(358) = 18.2, p < .001$; Simulated and Human vs Human Only: $t(358) = 16.1, p < .001$). This demonstrates that simulated data is beneficial in addition to human data in predicting real human preferences.

To assess overall accuracy at predicting human preferences from fixations alone, we compared LSTMs trained on human and simulated data to control MLPs trained on hand-designed features. As with the simulated data case, the best performing control model for predicting human preferences from fixations was provided the current fixation identity, the sum of fixations on each item up to that time-point, and the proportion of fixations up to that time-point (Appendix Fig. 4B). This model was outperformed by LSTMs trained on simulated and human data (Fig. 3A; $t(358) = 37.3, p < .001$). As in the simulated data case, LSTMs trained on simulated and human data, utilizing both information about fixations and which item was chosen, performed better than a model that only used information about which item was chosen (Fig. 3B; $t(358) = 12.52, p < .001$). This demonstrates that, under this approach, using fixation data to predict preferences confers a slight benefit beyond simply predicting which item will be chosen.

## 4. Discussion

Cognitive models, which define the relationships between an individual's latent preferences and their behavior, offer a tremendous opportunity to infer the hidden variables that guide an individual's choice. However, standard approaches to performing probabilistic inference with such models are computationally prohibitive for practical applications. Here, we have proposed and implemented a new approach for using neural networks to perform inference in such cognitive models, which can make inference computationally feasible for online applications. In addition to demonstrating that neural networks can perform inference of latent preferences in such models, we have also demonstrated that simulating data from such models can make up for limited human data in training neural networks to infer real human preferences from behavior.

Overall, this approach is likely limited by the extent to which cognitive models can capture idiosyncratic features of the relationship between human preferences and behavior. In future work, we can improve this approach by identifying and understanding discrepancies between model generated datasets and real human fixation data. Identifying such discrepancies may enable the generation of new generative models of fixations. These models may relax the strong optimality assumptions of the model we currently use, but may in turn produce fixation data that is more useful for training neural networks for predicting preferences.

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

## Appendix A. Supplementary Figures

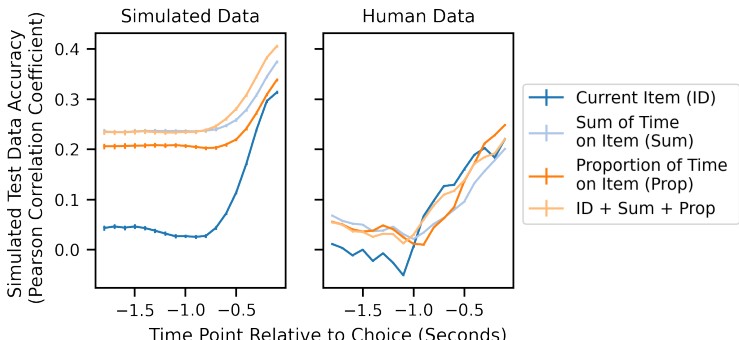

Figure 4: Performance of non-sequential control models. Control models are all multi-layer perceptrons (MLPs) which take in pre-defined features and output predictions of each item's utility. The Current Item (ID) model maps the current fixated item (represented as a length-3 one-hot vector) to a prediction of item utilities. Sum of Time on Item (Sum) and Proportion of Time on Item (Prop) map either the cumulative sum or cumulative proportion of time thus-far into the trial fixated on each item to a prediction of the item utilities. ID + Sum + Prop stitches these different representations together into a length-9 vector. A) Performance when trained and tested on simulated data. B) Performance when trained on simulated and human data and tested on human data. A,B) In both training and testing settings, ID + Sum + Prop achieved the best aggregate prediction accuracy across time-points, with aggregate $r = .27$ for Simulated data and $r = .10$ for human data.

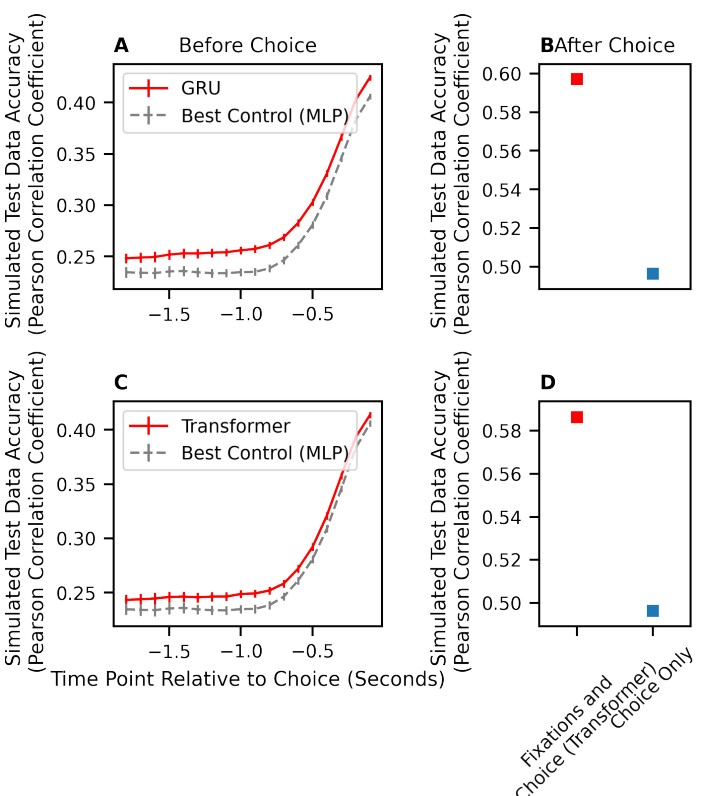

Figure 5: Results of training model on simulated data and testing on held-out simulated data for GRU and Transformer networks. Plot corresponds to Fig. 1, but with GRU (A,B) and Transformer (C,D) networks. A) GRU networks outperform control models at predicting from fixations prior to choice ($t(358) = 19.4, p < .001$). B) GRU networks utilizing fixation and choice data outperform predicting preferences from choice alone ($t(358) = 15.3, p < .001$). C) Transformer networks outperform control models at predicting from fixations prior to choice ($t(358) = 78.7, p < .001$). D) Transformer networks utilizing fixation and choice data outperform predicting preferences from choice alone ($t(358) = 37.3, p < .001$).

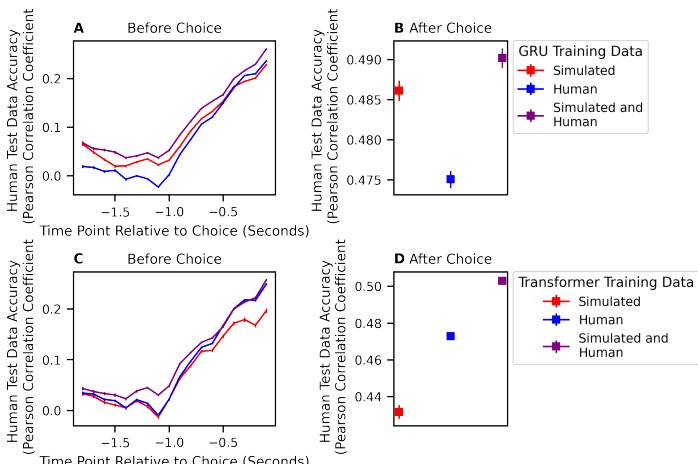

Figure 6: Testing on human data under different training regimes for GRU and Transformer networks. Plot corresponds to Fig. 2 but with GRU (A,B) and Transformer (C,D) networks. GRUs trained on simulated and human data outperforms GRUs trained on either simulated or human data only both prior to choice (A; Simulated and Human vs Simulated Only: $t(358) = 15.9, p < .001$; Simulated and Human vs Human Only: $t(358) = 28.4, p < .001$) and following choice (B; Simulated and Human vs Simulated Only: $t(358) = 5.0, p < .001$; Simulated and Human vs Human Only: $t(358) = 19.4, p < .001$). Transformers trained on simulated and human data outperforms Transformers trained on either simulated or human data only both prior to choice (C; Simulated and Human vs Simulated Only: $t(358) = 21.0, p < .001$; Simulated and Human vs Human Only: $t(358) = 11.2, p < .001$) and following choice (D; Simulated and Human vs Simulated Only: $t(358) = 38.0, p < .001$; Simulated and Human vs Human Only: $t(358) = 40.8, p < .001$).

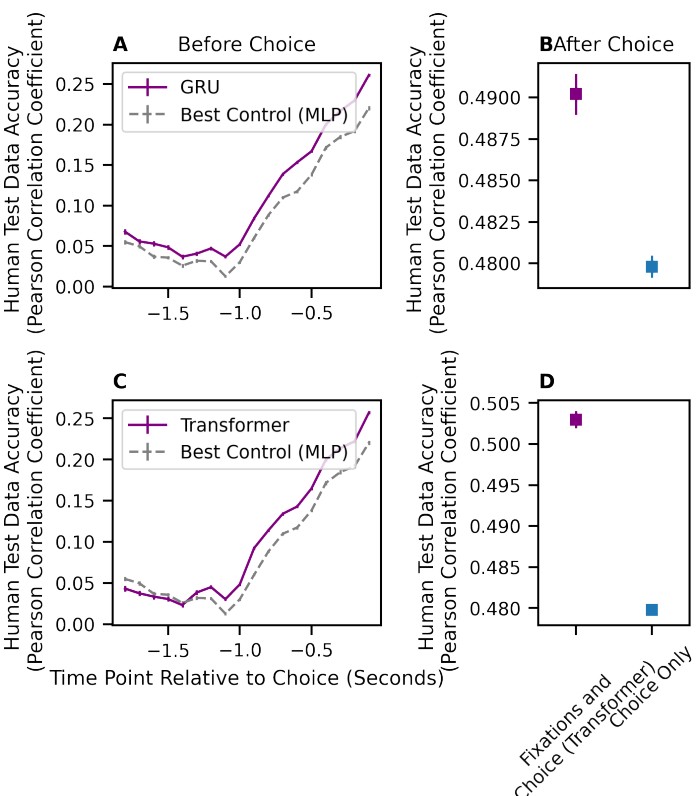

Figure 7: Results of training model on simulated and human data and testing on held-out human data for GRU and Transformer networks. Plot corresponds to Fig. 3, but with GRU (A,B) and Transformer (C,D) networks. A) GRU networks outperform control models at predicting from fixations prior to choice ($t(358) = 42.3, p < .001$). B) GRU networks utilizing fixation and choice data outperform predicting preferences from choice alone ($t(358) = 12.5, p < .001$). C) Transformer networks outperform control models at predicting from fixations prior to choice ($t(358) = 38.7, p < .001$). D) Transformer networks utilizing fixation and choice data outperform predicting preferences from choice alone ($t(358) = 31.8, p < .001$)

