# OpenReview forum: "Inverting cognitive models with machine learning to infer preferences from fixations"
_NeurIPS.cc/2023/Workshop/Gaze_Meets_ML — Gaze Meets ML 2023 Poster_

### Official Review · Reviewer_zWcf · 2023-10-12
**Interesting approach to training neural networks on tasks with low data availability**

**Rating:** 6
**Confidence:** 3

**Review:**

The paper proposes a new method to infer an individual’s preferences from their observable behavior (i.e., scanpaths). Previous approaches were limited by the availability of annotated data. The authors propose to first generate large quantities of synthetic data using a cognitive model. Afterward, they train a neural network to predict preferences on the data generated by the cognitive model. They show that combining synthetic and real behavioral data increases the performance over only using real data.

While the paper is easy to understand and the proposed methodology is interesting I would have preferred a more extensive description of the data generation process. "Simulated data was generated using the model described in Anonymous". Here it would have been helpful to just refer to the paper in the third person, which would not have broken the double-blind review.

---

### Official Review · Reviewer_bZ5f · 2023-10-22
**Solid paper demonstrating usage of machine learning to invert a cognitive model on eye movement when making choices**

**Rating:** 8
**Confidence:** 3

**Review:**

The authors explore whether utilities of certain choices (e.g., between three snacks) can be efficiently estimated using machine learning models based on eye tracking data. Due to limited data availability, they employ eye gaze data simulated from known random utilities using cognitive models. A neural network (LSTM, GRU, Transformer) is trained to reconstruct these utilities based on the fixations and/or the final choice (e.g., which snack was selected).

Strengths:
* Both training/evaluating only on synthetic data and fine-tuning/evaluating on human data are evaluated, allowing for a reasonable estimate of the synthetic-to-real distribution difference.
* Reported accuracy on human data demonstrates that pre-training with synthetic data is beneficial. They also demonstrate that eye gaze in itself is beneficial compared to only knowing the final choice.
* A variety of different neural network architectures (LSTM, GRU, Transformer) are tested and employed. Differences seem to be minor and thus GRU and Transformer are moved to the appendix, but it's good to have these results.

Weaknesses:
* The rationale and applicability of the employed cognitive model cannot be determined, as it is an anonymous reference. The authors should have included the full reference. Used in third person this does not typically affect anonymity.
* Model inputs and outputs are clearly described in text but not easy to understand upon first reading the paper. Readability of the section 2.3 in particular would benefit from a graphical representation of the models described therein.

---

### Official Review · Reviewer_afYK · 2023-10-24
**Paper shows an innovative idea to solve an important problem but lacks basic considerations and has low contribution.**

**Rating:** 4
**Confidence:** 4

**Review:**

Originality: The methods used are innovative and forward-thinking.

Quality of the paper: The paper needs more basic (gaze) considerations, which diminishes its contribution.

Significance: While the approach has great potential, its contribution is pretty low.


Specific comments to the authors:

Introduction:

1. The exploration of individual eye-tracking patterns is a promising area of research, and your engagement with this topic is commendable. To fortify the study's foundation, knowing the number of participants behind the 2,965 trials would be helpful. This can offer insights into data breadth and the potential for varied outcomes.
Human Data:

2. A deeper dive into trial durations and the accuracy of the eye tracker would provide readers with a clearer understanding of the methodologies employed. Also, elaborating on the event detection method could help you understand better. While the Gaussian distribution addition is innovative, delineating its impact with other data variables might clarify its efficacy.
Simulated Data:

3. Your commitment to producing 1.8 million new sample trials showcases your dedication to the research. However, with the inherent challenges in eye-tracking data, the quality of input is paramount. Clarifying whether the goal is individual-specific relationships or a more generalized cross-subject model is essential. If the latter, a distinct split of participants between training and validation sets would strengthen the model's validity and eliminate potential biases.
Results:

4. Presenting the correlation between your method and the best control is an interesting choice. Given the nature of Pearson's correlation, underlying relationships might not be captured. Comparing with the ground truth could provide alternative insights.

5. The predictive outcomes around the time point relative to choice "0" warrant further investigation. Delving deeper into this pivotal moment could enhance the model's overall predictability. The accuracy plots provide a glimpse into the potential of integrating fixations and choice data. However, while innovative, the interplay between simulated and human data raises questions about data quality and its impact on outcomes.

Summary:
I think your endeavor to combine fixations with choice prediction is important. However, pressing concerns need to be addressed to solidify the study's contributions. The nuances of individual variability, the attributes of the eye-tracking device, and the quality of simulated data are pivotal points that require attention. While the paper offers valuable insights and demonstrates a forward-thinking approach, it requires substantial revisions before it can be accepted in its current form. Your dedication to the research is evident, and with a few adjustments, this work has the potential to be groundbreaking in the domain.

---

### Meta-Review · Area_Chair_9edy · 2023-10-25

**Recommendation:** Accept (Poster)
**Confidence:** 5

**Metareview:**

The authors have introduced a new method to estimate human cognitive choices by combining a limited set of real gaze data with a large amount of simulated data from cognitive models. All the reviewers appreciated this work. They particularly liked integrating synthetic and real data to understand individual people's choices using eye tracking.

However, the reviewers criticized the need for more information about the data set creation, the distribution of the human-collected gaze data, and some of the details of the results presented.

De-anonymizing the dataset collection and adding crucial details on the dataset collection will help address the reviewers questions.

---

### Decision · Program_Chairs · 2023-10-26

Accept (Poster)